# Drug Repurposing for Non-Alcoholic Fatty Liver Disease by Analyzing Networks Among Drugs, Diseases, and Genes

**DOI:** 10.3390/metabo15040255

**Published:** 2025-04-09

**Authors:** Md. Altaf-Ul-Amin, Ahmad Kamal Nasution, Rumman Mahfujul Islam, Pei Gao, Naoaki Ono, Shigehiko Kanaya

**Affiliations:** Computational Systems Biology Lab, Graduate School of Science and Technology, Nara Institute of Science and Technology, Nara 630-0101, Japan; kamal.nasution@naist.ac.jp (A.K.N.); rumman.mahfujul_islam.ro7@is.naist.jp (R.M.I.); gao.pei.gi3@is.naist.jp (P.G.); nono@is.naist.jp (N.O.); skanaya@is.naist.jp (S.K.)

**Keywords:** BiClusO algorithm, disease similarity, drug repurposing, network analysis, non-alcoholic fatty liver disease (NAFLD)

## Abstract

Background/Objectives: Drug development for complex diseases such as NAFLD is often lengthy and expensive. Drug repurposing, the process of finding new therapeutic uses for existing drugs, presents a promising alternative to traditional approaches. This study aims to identify potential repurposed drugs for NAFLD by leveraging disease–disease relationships and drug–target data from the BioSNAP database. Methods: A bipartite network was constructed between drugs and their target genes, followed by the application of the BiClusO bi-clustering algorithm to identify high-density clusters. Clusters with significant associations with NAFLD risk genes were considered to predict potential drug candidates. Another set of candidates was determined based on disease similarity. Results: A novel ranking methodology was developed to evaluate and prioritize these candidates, supported by a comprehensive literature review of their effectiveness in NAFLD treatment. Conclusions: This research demonstrates the potential of drug repurposing to accelerate the development of therapies for NAFLD, offering valuable insights into novel treatment strategies for complex diseases.

## 1. Introduction

Drug repurposing, also known as drug reprofiling or re-tasking, is a strategy used to identify new applications for existing approved or investigational drugs beyond their initial medical indications [1]. The development of new pharmaceutical compounds is both costly and time-consuming, despite the pressing demand for new treatments. A 2014 workshop estimated that the process of successfully bringing a novel drug to market can take around a decade and require an investment of up to USD 1 billion [2]. Within the biopharmaceutical industry, companies have continuously sought ways to enhance productivity by adopting innovative drug discovery approaches, one of which is drug repurposing. This method offers a promising alternative by re-evaluating existing drugs for potential use in treating other diseases. Additionally, it provides significant advantages by reducing both time and financial burdens while also assessing the efficacy of currently available treatments. Drug repurposing has emerged as a pivotal strategy in pharmaceutical research, aiming to identify novel therapeutic applications for existing drugs. This approach significantly expedites drug development by leveraging established safety, efficacy, and pharmacokinetic data, thereby lowering the time, risk, and cost compared to de novo drug discovery. In recent years, it has gained particular traction in tackling complex diseases, such as non-alcoholic fatty liver disease (NAFLD), by exploiting known interactions among drugs, genes, and pathways. As a result, drug repurposing enables clinicians and researchers to more swiftly adapt existing treatments for new clinical indications, ultimately improving patient care while conserving resources [3].

Non-alcoholic fatty liver disease (NAFLD) has emerged as a critical global health challenge, driven by the increasing prevalence of obesity and type 2 diabetes [4]. As NAFLD progresses, it can lead to more severe conditions such as cirrhosis and hepatocellular carcinoma, significantly burdening both healthcare systems and patients [5]. Despite the escalating impact, treatment options for NAFLD remain limited, and there are currently no FDA-approved drugs specifically for this condition [6]. Lifestyle modifications, such as those to diet and exercise, are the primary management strategies, yet these interventions are usually insufficient, particularly in advanced stages of the disease [7]. This unmet need underscores the urgency for developing effective pharmacological treatments. Drug repurposing has emerged as a promising strategy to accelerate drug discovery by identifying new therapeutic uses for existing drugs, which can significantly reduce the time and cost associated with drug development [8].

Drugs may affect the rate at which existing biological functions proceed, and drugs usually do not change the basic nature of these functions or create new functions. Drug receptors are usually proteins, i.e., the products of genes that reveal polymorphisms. Drug–gene interaction (DGI) is an association between a drug and a gene that may affect a patient’s response to drug treatment. Therefore, comprehensive DGI associations contain system-level clues regarding disease mechanisms and can be utilized for drug repurposing. Scientists have developed DGI databases based on various published scientific studies that can be used to examine the functional modules of specific drug sets and their target interactions.

Furthermore, disease–gene association (DGA) data contain information on disease-associated genes. Information about the genes and variants involved in human diseases can be used for the investigation of the molecular mechanisms of diseases and their comorbidities, the analysis of the properties of disease–gene relations, and the generation of hypotheses on drug therapeutic action and adverse effects [9]. Disease–gene association (DGA) can be used for calculating disease–disease similarity, which can lead to a particular disease being connected to potential drugs with the usage of drug–gene interactions. Analyzing disease–disease relationships plays an important role in understanding disease mechanisms and finding alternative uses for a drug [10]. The similarity between two diseases can be computed as a function of the associated genes or, alternately, the biological processes related to them [11]. Previous research has utilized DGI and DGA data for drug repurposing for inflammatory bowel disease [12].

In this study, we applied a system-level computational strategy to explore drug repurposing opportunities for NAFLD by integrating drug–gene interactions and disease–gene associations.

## 2. Materials and Methods

In this work, we applied two different approaches for repurposing drugs for NAFLD. One of the approaches is based on the bi-clustering of a drug–gene interaction network and NAFLD risk genes. The other approach is based on disease similarity calculation and identifying drugs relevant to diseases that are similar to NAFLD.

### 2.1. Data

Data for this study were collected and processed from multiple publicly available databases, including BioSNAP [13], GWAS [14], CTD [15], and DisGeNET [9]. Below, we discuss the major datasets collected for this study.

NAFLD risk genes: NAFLD risk genes were downloaded from three databases: the Comparative Toxicogenomics Database (CTD), DisGeNET, and GWAS. The CTD contains many genes, so we selected around 2.5% high-relevance genes based on inference scores [11]. The statistics of the downloaded and selected risk genes are shown in Table 1.

Drug–gene interaction data: We downloaded drug–gene interaction (DGI) data from the BioSNAP database (https://snap.stanford.edu/biodata/datasets/10002/10002-ChG-Miner.html, accessed on 15 January 2025). These data consist of 15,138 DGIs among 5017 unique drugs and 2324 unique genes. In this dataset, the maximum degree of a drug is 147, the minimum degree is 1, and the average degree is 3.017. Only 1 drug called NADH is connected to 147 genes, and 2986 drugs are connected to only 1 gene. Lepirudin is an example drug which is related to only 1 gene.

Disease–gene association data: We downloaded disease–gene association (DGA) data which have 21,357 entries involving 519 unique diseases and 7294 unique genes from the BioSNAP database (https://snap.stanford.edu/biodata/datasets/10012/10012-DG-AssocMiner.html, accessed on 15 January 2025). In this dataset, the maximum degree of a disease is 485, the minimum degree is 10, and the average degree is 41.150. Only 1 disease is related to 485 genes, and 41 diseases are connected to 10 genes. For example, *Adenomatous Polyposis Coli* is connected to 10 genes, and Prostatic Neoplasms are connected to 485 genes.

### 2.2. Methods

In this study, we employed two different methods for drug repurposing for NAFLD. One of the methods corresponds to the bi-clustering of drug–gene interaction data and incorporating NAFLD risk genes. The other method is based on disease similarity calculated using disease–gene association data. Below, we discuss these two methods.

#### 2.2.1. Drug Selection by Bi-Clustering

The workflow of this method is shown in Figure 1a. The drug–gene interaction data can be represented as a bipartite network. We applied the BiClusO [16,17,18] algorithm to these data to generate densely connected bi-clusters. Then, the bi-clusters that are rich in NAFLD risk genes were identified using enrichment fold calculation. The drugs included in such clusters were considered effective against NAFLD.

#### 2.2.2. Drug Selection Based on Disease Similarity

The workflow of this method is shown in Figure 1b. In this method, we calculated the similarity between two diseases based on their target genes. First, we collected a list of diseases and their associated genes from the BioSNAP database. The similarity score for each disease was calculated by comparing the genes associated with NAFLD to the genes associated with each of the other diseases. For this, the ‘DOSE’ package of R was used, and by employing the ‘clusterSim’ function, the disease similarity was calculated. The matching score ranges between 0 and 1, where 1 indicates the maximum similarity. Additionally, we merged the DGI dataset and the DGA dataset and created a new network named disease–gene–drug association (DGDA). After that, based on a selected number of diseases highly similar to NAFLD, we determined potential drugs for NAFLD utilizing the DGDA network.

### 2.3. Ranking and Final Selection

After deriving potential drugs for NAFLD by applying two different approaches as described above, we took the intersection of them as promising drugs. Thus, support from two different approaches strengthened the accuracy of our prediction. We further ranked them using two metrics called the gene association score (GAS) and disease coverage score (DCS).

## 3. Results

The primary objective of this research was to construct a multidimensional data network linking drugs, genes, and diseases to evaluate their potential for drug repurposing for non-alcoholic fatty liver disease (NAFLD). Below, we discuss the results obtained from two different methods and how we made the final prediction list by using some ranking criteria.

### 3.1. Screening Potential Drugs by Bi-Clustering

We utilized the BiclusO algorithm to perform bi-clustering on drug–gene interaction data. For this purpose, we used the software tool DPClusSBO 1.2 [17]. The basic theory of BiClusO is to convert a two-dimensional problem to a one-dimensional one by data folding, solve it using a one-dimensional algorithm, and then unfold it again. Thus, the BiClusO algorithm first converts a bipartite graph to a simple graph by taking any node set and measuring the association between node pairs using a relation number and Tanimoto coefficient, then performs simple graph clustering using the polynomial-time heuristic algorithm DPClusO we developed before [17]. Finally, the attachment of the nodes from the second set creates each bi-cluster.

The main parameter values utilized for this algorithm are listed in Table 2. These parameter values are balanced and suitable for a sparse bipartite graph as our DGI network. Using 5017 drugs and their corresponding 2324 genes, we performed bi-clustering analysis. In total, 607 bi-clusters were identified involving drugs and genes. The resulting clusters were integrated with NAFLD risk gene data to identify potential drug candidates.

To assess the richness of NAFLD genes in the identified bi-clusters, we calculated the enrichment fold compared to the expectation for each cluster. The enrichment fold was calculated according to the following equation:
$$\mathrm{Enrichment}\;\mathrm{Fold}=\frac{L/M}{K/N}$$ where N = total number of elements in the population; K = number of elements in the population belonging to the category of interest; M = number of elements in the selected subset; L = number of elements in the selected subset belonging to the category of interest.

The distribution of clusters concerning enrichment fold is illustrated in Figure 2. We empirically selected a threshold of 2.75, allowing for the selection of the top value clusters. Subsequently, the clusters exhibiting an enrichment fold greater than this value were screened. A total of 47 clusters met this criterion. Figure 3 shows an example bi-cluster where 7 out of 11 genes are NAFLD genes. The enrichment fold for this cluster is 2.75. Within these selected clusters, 230 drugs were identified as potential candidates for non-alcoholic fatty liver disease (NAFLD) treatment. Figure 2The distribution of clusters in the context of enrichment fold.
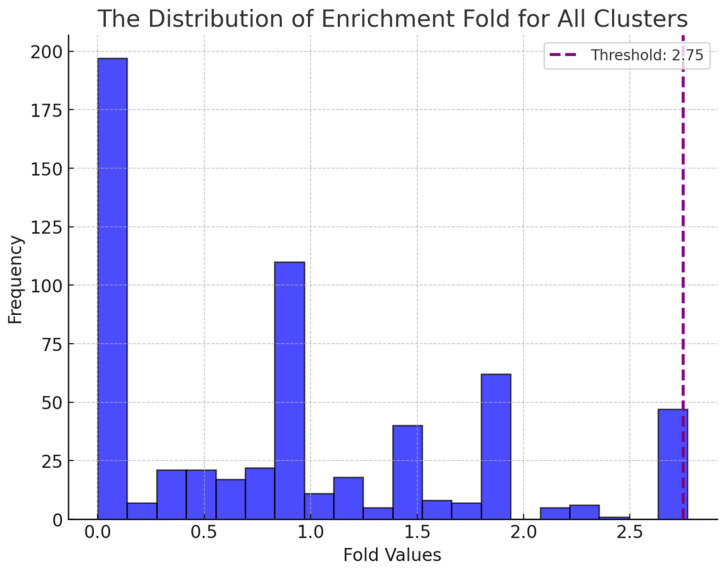

Figure 3Bipartite graph of drug–gene interactions for top-performing cluster.
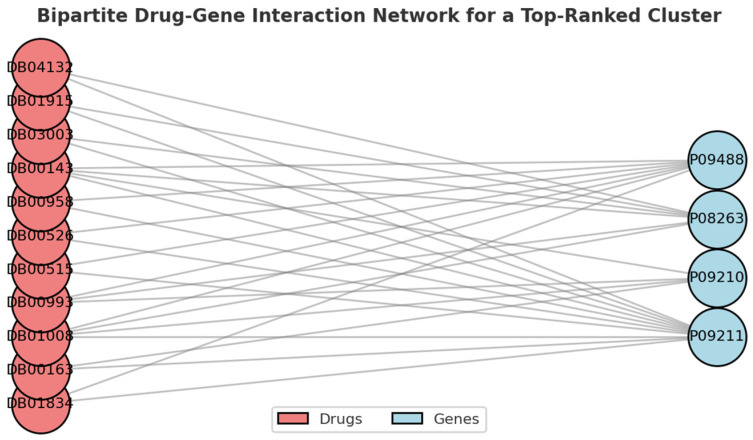


### 3.2. Screening Potential Drugs by Disease Similarity

Following the identification of potential drug candidates through bi-clustering, we further calculated disease similarity to explore potential relationships between NAFLD and other diseases using gene–disease association data from the BioSNAP database. The rationale behind this step is to leverage the shared genetic and pathological features of diseases to identify potential therapeutic targets. By utilizing the NAFLD gene set as a reference, we computed similarity scores for a range of diseases using the DOSE package in R [19], as shown in Figure 4. These scores indicate the genetic similarity of diseases to NAFLD, aiding in prioritization for further study.

The results of the similarity analysis revealed that Drug Allergy, Atherosclerosis, and Non-Insulin-Dependent Diabetes Mellitus (NIDDM) have particularly high similarity scores with NAFLD. These diseases are characterized by similar disruptions in lipid metabolism and immune responses, making them prime candidates for comorbidity studies and potential drug repurposing. Additionally, diseases such as Neuralgia and Papillary Thyroid Carcinoma, which have not been extensively studied in relation to NAFLD, were also found to share significant genetic similarities. This highlights the value of integrating disease similarity into the drug discovery process.

Additionally, we merged the DGI dataset and the DGA dataset and created a tripartite network named disease–gene–drug association (DGDA). Figure 5 shows a typical network representing drug–gene–disease association. After that, based on varying similarity scores (by keeping a varying number of disease nodes in the DGDA network), we determined how many drugs were associated with the diseases that are similar to NAFLD. The relation between associated drugs and disease similarity is shown in Figure 6. From Figure 6, we empirically selected 0.95 as a threshold disease similarity value for this study. Corresponding to the 0.95 similarity score, there are 3885 drugs and 88 diseases.

### 3.3. Final Selection

To ensure the robust identification of drug candidates for NAFLD treatment, we employed a multi-step process integrating both clustering and disease similarity analyses. Initially, we used the BiclusO algorithm to identify the clusters of drugs and genes linked to NAFLD. We identified 230 drugs based on the bi-clustering approach. And 3885 drugs associated with these 88 diseases were identified based on the disease similarity approach (Figure 7). Finally, we focused on the intersection of these two predicted sets of drugs. Thus, we found 92 potential drug candidates which were selected by both methods, offering promising therapeutic avenues for NAFLD.

To evaluate these drugs, we developed two scoring systems: the **d****isease coverage score (DCS)** and the **gene association score (GAS)**. The DCS reflects the number of diseases similar to NAFLD that are associated with a drug. The **gene association score (GAS)** is used to calculate the number of genes associated with a drug.

From the DGDA data, we extracted the drug vs. disease bipartite relationships involving 88 diseases that are very similar to NAFLD and 92 selected drugs. Drug and disease pairs are connected when there is at least one path linking a drug to a disease through a gene in the network shown in Figure 5. This extraction can be conducted by multiplying a relevant part of the drug–gene interaction (DGI) data matrix with the disease–gene association (DGA) data matrix. Let this drug vs. disease bipartite relationship be represented by a binary matrix M_DD_. This score quantifies the extent to which a drug is associated with multiple diseases based on the given drug–disease interaction matrix. Then, the disease coverage score DCS_i_ for the *i*th drug is defined as follows:$$DCSi=\sum_{j=1}^{N\;}M_{\mathrm{DD}}\left(ij\right),\;\;\;\;\;N=88$$ where M_DD_(ij): binary value indicating the association between the *i*th drug and *j*th disease; N = 88 represents the number of diseases with a similarity greater than 0.95 to NAFLD.

The GAS is used to measure the number of unique genes linked to each drug in the drug–gene association data, offering insights into the drug’s genetic significance. Drugs with higher GAS values are prioritized for their broader genetic relevance. Let the drug–gene bipartite relationships be represented by the matrix M_DG_. Then, the formula for the GAS of the *i*th drug is as follows: $$GASi=\sum_{j=1}^{K\;}M_{\mathrm{DG}}\left(ij\right),\;\;\;\;\;K=2324$$ where M_DG_(ij): binary value indicating the association between the *i*th drug and *j*th disease; K = 2324 represents the number of diseases.

By combining both the DCS and GAS into a weighted total score, we were able to rank the potential drugs and identify the top candidates. The total score is calculated as follows:*Total Score = α × DCS + β × GAS*.

Using α = 1 and β = 1 allows for the equal weighting of both scores to ensure that both factors contribute equally without bias. This selection was based on empirical observations, providing a balanced and interpretable result in this study. We calculated the total score for 92 selected drugs, and the 15 top-ranking drugs are shown in Table 3. These drugs are linked to promising evidence for NAFLD treatment. These drugs have system-level associations with NAFLD risk genes and diseases similar to NAFLD and high DCS and GAS values (Table 3). Thus, through a systematic integration of clustering, disease similarity, and scoring mechanisms, we confidently identified a selected group of drugs that warrant further experimental validation for their efficacy in treating NAFLD. Out of 92 predicted drugs, we surveyed the literature to identify the top 15 drugs and provide a discussion of them here.

### 3.4. Validation of Potential Drugs

We conducted a comprehensive manual literature review to substantiate our selection of the top 15 repurposed drug candidates for non-alcoholic fatty liver disease (NAFLD), as presented in Table 3.

**Andrographolide** (HMPL-004) is a botanical product extracted from an herb that occurs naturally in China. The herb has an extensive history of use in TCM for the treatment of upper respiratory tract infections and other inflammatory and infectious diseases. This drug suppresses the expression of inflammation cytokines including TNF-α, IL-1β, and IL-6, which are all considered pro-inflammatory cytokines that are significantly linked to the development and progression of non-alcoholic fatty liver disease (NAFLD), meaning that elevated levels of these cytokines in the blood can indicate an increased risk of NAFLD. These cytokines, particularly TNF-α, play a key role in activating inflammatory pathways within the liver, contributing to the damage of hepatocytes (liver cells) and promoting the development of NAFLD, especially when progressing to the more severe stage known as non-alcoholic steatohepatitis (NASH). Thus, there is an association of inflammatory cytokines with non-alcoholic fatty liver disease [20].

**Clozapine** is a drug used to treat a severe form of a psychiatric condition known as schizophrenia. NAFLD/NASH is highly comorbid with psychiatric disorders; hence, Clozapine might indirectly help treat NAFLD. But it has been reported that this drug can worsen Glucose Intolerance and non-alcoholic fatty liver disease [21]. Therefore, it can be said that our method selected Clozapine because it can substantially affect pathways involving liver, though this may be in a negative sense. This shows the need for the experimental validation of the computational results before clinical application.

**VX-703** inhibits p38 MAP kinase, which effectively inhibits pro-inflammatory cytokines including TNF-α, IL-1β, and IL-6. The liver regulates glucose and lipid metabolism, and these processes become dysfunctional under pathophysiological conditions such as obesity, type 2 diabetes mellitus (T2DM), and non-alcoholic fatty liver disease (NAFLD). Stress responses activate hepatic MAPKs, and this is thought to impair insulin action and lipid metabolism. Thus, p38 MAP kinase activation plays a role in the pathogenesis of fatty liver and NASH. Therefore, as an inhibitor of p38, VX-703 is likely to be an effective drug for NAFLD [22,23].

**Dilmapimod** has been used in trials studying the treatment and diagnosis of Nerve Trauma, inflammation, Pain, Neuropathy, arthritis, Rheumatoid arthritis, and Coronary Heart Disease, among others. Like **VX-703**, Dilmapimod (SB-681323) is a p38 MAP kinase inhibitor. So as discussed above, it might be good for NAFLD.

**Vitamin E**’s antioxidant activity helps combat oxidative stress within the liver, which is a key factor in NAFLD progression [24]. Some research suggests that vitamin E may positively impact liver histology by reducing steatosis (fat accumulation) and inflammation in patients with NAFLD [25]. Vitamin E is often recommended as an adjunct therapy alongside lifestyle changes like weight loss and dietary modifications for managing NAFLD [26].

**YSIL6** is an inhibitor of interleukin-6 (IL-6). IL-6 inhibitors are drugs that block the activity of IL-6, a protein that plays a role in inflammation. YSIL6 works by inhibiting TNF-α and IL-6 production in T-cells and macrophages. So, its action against NAFLD is similar to that of Andrographolide as discussed above.

**Talmapimod** was the first first-generation oral p38 MAP kinase inhibitor developed. It has been shown to be effective in curing inflammatory diseases. So as discussed above in the case of VX-703, it may be used to treat NAFLD.

**Dehydroepiandrosterone/prasterone** (DHEA) is used to correct DHEA deficiency due to adrenal insufficiency or old age, as a component of menopausal hormone therapy. It has been reported that DHEA prevents non-alcoholic steatohepatitis by activating the AMPK pathway mediated by GPR30 [27].

**Indomethacin** is a Non-Steroidal Anti-Inflammatory Agent. It has been reported that Fish oil and indomethacin in combination potently reduce dyslipidemia and hepatic steatosis in LDLR [28].

**Propofol** is a medication used as an anesthetic in surgery. Propofol is generally considered a relatively safe anesthetic option for patients with non-alcoholic fatty liver disease (NAFLD). Some studies suggest that propofol may have antioxidant effects that could potentially protect against liver damage [29].

**Sulfasalazine** is a medication used to treat some inflammatory bowel diseases and types of arthritis. Studies have shown that Sulfasalazine may inhibit the production of proteins responsible for scar tissue formation in the liver, which could potentially help in treating liver fibrosis associated with advanced NAFLD (https://www.medicalnewstoday.com/articles/52735#1) (accessed on 14 February 2025).

**Melatonin** is a naturally occurring hormone that promotes sleep. Melatonin may help improve non-alcoholic fatty liver disease (NAFLD). Melatonin is a natural hormone that may help with inflammation, oxidative stress, and liver enzymes [30]. One study mentioned that melatonin improved NAFLD in patients who took 6 mg per day for 12 weeks [31].

**Menadione (K3)** is converted to menaquinone (vitamin K2) in the liver. Another study reported that vitamin K2 protects mice against non-alcoholic fatty liver disease induced by a high-fat diet [32].

**As for ATP**, maintaining adequate levels of ATP (Adenosine Triphosphate) could be beneficial for managing NAFLD (non-alcoholic fatty liver disease) as low ATP levels are often observed in the liver of NAFLD patients due to mitochondrial dysfunction [33]. High fructose intake can significantly deplete hepatic ATP levels, potentially contributing to NAFLD development. Therefore, ATP supplements can hinder NAFLD development [34].

**Regarding Mesalazine,** studies have shown comorbidities between inflammatory bowel disease (IBD) and NAFLD, and since Mesalazine is an anti-inflammatory medication used to treat IBD, it could theoretically help manage inflammation associated with NAFLD as well [35,36].

## 4. Discussion

This study highlights the potential of drug repurposing for NAFLD using computational selection. Several identified candidates, including Andrographolide, VX-703, and Dilmapimod, inhibit key pro-inflammatory cytokines (TNF-α, IL-1β, IL-6), reducing hepatic inflammation. Antioxidants like vitamin E and melatonin further mitigate oxidative stress, a major driver of NAFLD.

While drug repurposing primarily aims to identify new therapeutic applications, it is equally important to recognize cases where existing drugs may have unintended adverse effects. Our analysis identified Clozapine, an antipsychotic used to treat schizophrenia, as a potential candidate for NAFLD/NASH-related pathways. This finding highlights a crucial aspect of computational drug repurposing. While such approaches can uncover important drug–disease associations, their predictions must be cautiously interpreted. Negative findings, like the potential risks of Clozapine use in NAFLD, emphasize the need for rigorous experimental validation before clinical application. Identifying beneficial and harmful drug interactions helps refine repurposing strategies and improve patient safety in clinical decision making.

The BiclusO algorithm effectively identified promising candidates, demonstrating the power of computational methodologies. However, experimental validation remains crucial to confirm therapeutic potential, assess off-target effects, and integrate these candidates with existing NAFLD treatments. Future research should refine predictive models such as network-based approaches, machine learning algorithms, and text-mining techniques to enhance accuracy and clinical applicability. Furthermore, the approach presented here is highly generalizable. It could be extended to other diseases by adapting the drug–gene bipartite network and similarity measures to other diseases under a specific biological context. This versatility allows for the identification of novel drug candidates for other diseases, including cancer, cardiovascular disorders, and neurodegenerative conditions.

In this study, we utilized well-curated data; however, we acknowledge that the datasets were prepared several years ago. Therefore, the outcomes might improve if more comprehensive and noise-free data can be collected and utilized.

## 5. Conclusions

This study underscores the potential of drug repurposing as a strategic approach to the discovery of effective therapies for non-alcoholic fatty liver disease (NAFLD). By utilizing disease–disease relationships and drug–target interactions, we successfully identified a refined list of promising drug candidates with strong associations with NAFLD risk genes. The application of the BiClusO bi-clustering algorithm facilitated the identification of high-density drug–gene clusters, revealing critical insights into the potential therapeutic uses of existing drugs. The evaluation of the top-scoring drugs, including Andrographolide, VX-702, and vitamin E, among others, illustrates a compelling link to NAFLD treatment, supported by substantial evidence in the literature.

We repurposed some existing drugs for NAFLD using only computational methods. Therefore, further experiments are needed to assess their effectiveness in clinical settings. However, experimental validation poses challenges due to biological variability and potential discrepancies between computational predictions and real-world outcomes. Translating these findings into clinical applications requires further validation and refinement to ensure robustness and reliability. Despite these challenges, this method can also be applied for drug repurposing for other diseases, offering a systematic framework for identifying novel therapeutic opportunities.

## Figures and Tables

**Figure 1 metabolites-15-00255-f001:**
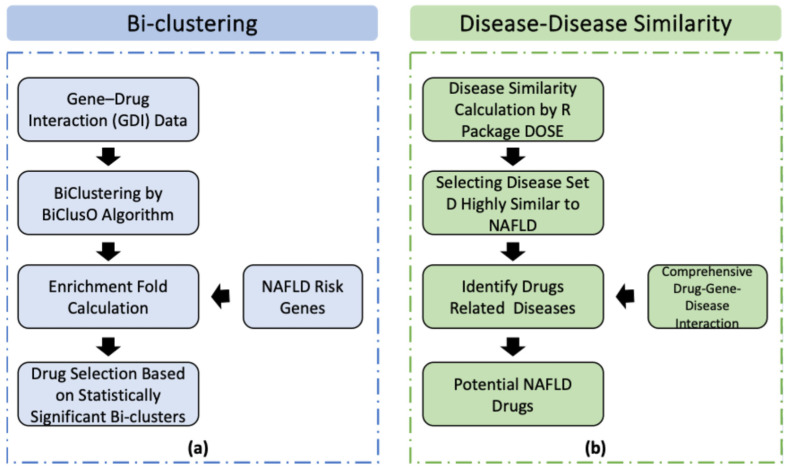
Workflow of two methods: bi-clustering (**a**) and disease–disease similarity (**b**) analysis.

**Figure 4 metabolites-15-00255-f004:**
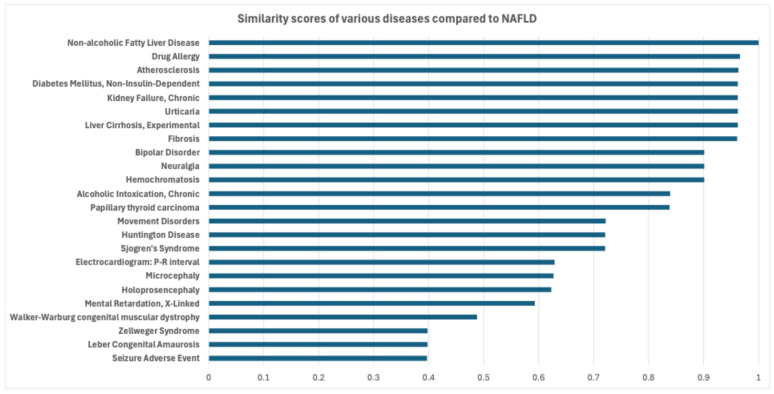
Similarity scores of various diseases compared to NAFLD.

**Figure 5 metabolites-15-00255-f005:**
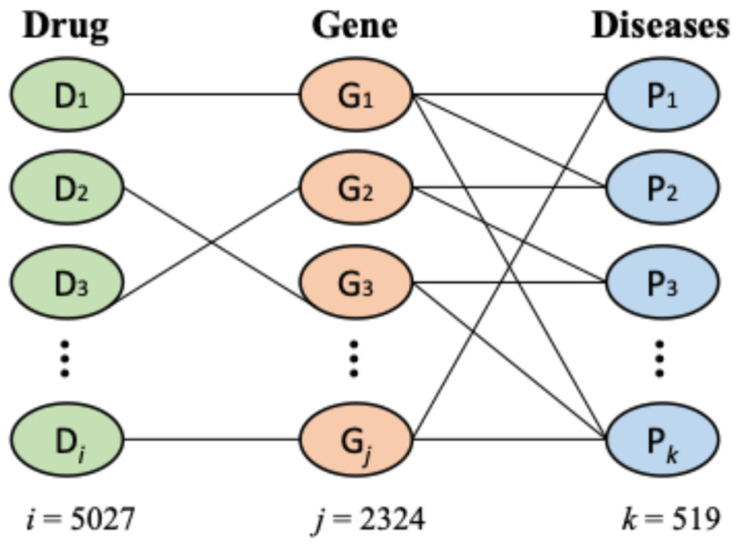
Network representation of drug–gene–disease association.

**Figure 6 metabolites-15-00255-f006:**
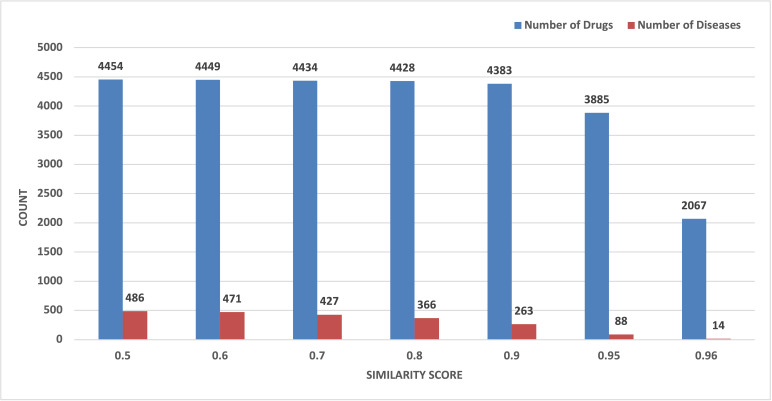
Drugs corresponding to diseases.

**Figure 7 metabolites-15-00255-f007:**
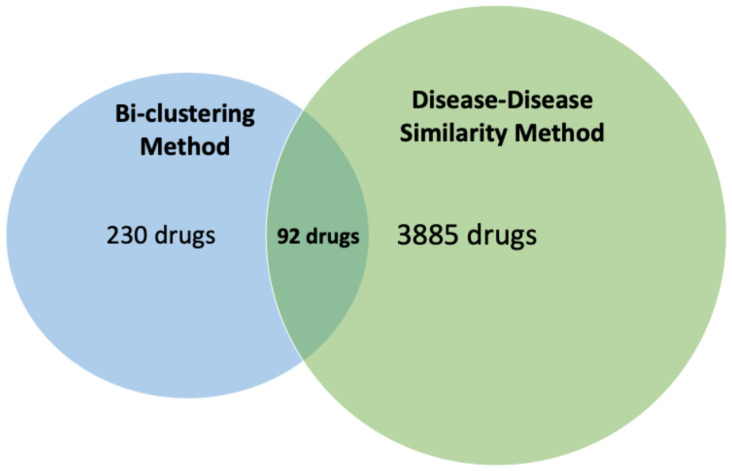
Illustrating the number of results from both methods and their intersection.

**Table 1 metabolites-15-00255-t001:** Overview of NAFLD risk genes from three databases.

Database	Initial Data	Processing Method	Final Data
CTD	37,193	Top 2.5% inference score	924
DisGeNET	1058	-	1058
GWAS	407	Uniprot ID conversion	312

**Table 2 metabolites-15-00255-t002:** Main parameters in BiClusO.

Parameter	Value	Description
Cluster property	0.5	Used for periphery tracking in simple graph clustering.
Cluster density	0.5	Minimum density for generated simple clusters.
Tanimoto coefficient	0.1	Used as a threshold to convert a bipartite graph to a simple graph.
Relation number	1	Used as a threshold to convert a bipartite graph to a simple graph.
Attachment probability	0.5	Threshold for finding the second set of nodes of a bi-cluster.

**Table 3 metabolites-15-00255-t003:** Potential drugs with their respective scores.

Potential Drug	DrugBank Id	DC Score	GA Score	Total Score
Andrographolide	DB05767	76	5	81
Clozapine	DB00363	42	38	80
VX-702	DB05470	76	4	80
Dilmapimod	DB05250	76	3	79
Vitamin E	DB00163	59	18	77
YSIL6	DB05017	75	2	77
Talmapimod	DB05412	68	4	72
Prasterone	DB01708	37	34	71
Indomethacin	DB00328	55	14	69
Propofol	DB00818	32	35	67
Sulfasalazine	DB00795	54	11	65
Melatonin	DB01065	48	17	65
Menadione	DB00170	44	20	64
ATP	DB00171	30	34	64
Mesalazine	DB00244	56	7	63

## Data Availability

The data and code used are available via GitHub https://github.com/kamalNasution/NAFLD_Paper_data, uploaded on 14 March 2025.

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
