# Peer review of "Drug Repurposing for Non-Alcoholic Fatty Liver Disease by Analyzing Networks Among Drugs, Diseases, and Genes"

_metabolites, 2025, doi:10.3390/metabo15040255_

Round 1
Reviewer 1 Report
Comments and Suggestions for Authors
Title: Drug Repurposing for Non-Alcoholic Fatty Liver Disease by Analysis of Networks Among Drugs, Diseases, and Genes
General Comments:The manuscript presents a comprehensive computational study on drug repurposing for Non-Alcoholic Fatty Liver Disease (NAFLD) by integrating drug-gene and disease-gene interaction data. The approach is methodically sound and offers potential therapeutic avenues. However, several issues regarding English grammar, typographical errors, clarity, and scientific novelty need to be addressed before the manuscript is suitable for publication.
Specific Comments:
- The phrase "drug development for complex diseases like non-alcoholic fatty liver disease (NAFLD)" is repetitive. Suggestion: "Drug development for complex diseases such as NAFLD..."
- Line 20: "supported by a thorough literature review of their effectiveness in NAFLD treatment" is awkward. Suggestion: "supported by a comprehensive literature review on their effectiveness in treating NAFLD."
- Line 123: "The flow of this method is shown in figure 1 (a)" should be "The workflow of this method is shown in Figure 1a."
- Line 214: "These similarity scores provide insights into how closely related different diseases are to NAFLD at the genetic level, thus enabling the prioritization of diseases for further study." This is overly long. Suggestion: "These scores indicate the genetic similarity of diseases to NAFLD, aiding in prioritization for further study."
- Table 2: The parameters listed need a brief description for better understanding.
- Line 283: The formula for DCS could be explained more clearly. Define all variables properly.
- Line 576: The reference to Clozapine as potentially worsening NAFLD could benefit from a brief discussion on the implications of negative drug repurposing findings.
- The use of a bipartite network combined with disease similarity is commendable. However, the study would benefit from a brief comparison with other computational approaches used for drug repurposing.
- Highlight how this approach could be generalized to other diseases.
- Section 2.2.1: Clarify the rationale behind selecting 2.75 as the threshold for enrichment fold.
- Section 2.3: Explain why equal weighting (alpha = 1, beta = 1) was chosen for the total score calculation.
- The conclusion effectively summarizes the findings but should briefly discuss potential challenges in experimental validation and clinical translation.
Overall comments:
- Revise for grammar, punctuation, and typographical consistency.
- Highlight the novelty and broader applicability of the methodology.
- Discuss practical challenges and future directions in drug repurposing.
- After implementing these suggestions, the manuscript will be better positioned for publication.
Minor corrections required.
Reviewer 2 Report
Comments and Suggestions for Authors
This manuscript presents a well-conceived computational approach to drug repurposing for NAFLD, employing two complementary strategies: biclustering of drug-gene networks using BiClusO, and disease similarity analysis based on gene associations. The integration of these methods and the development of a ranking system based on DCS and GAS metrics adds robustness to the predictions. Furthermore, the literature-based validation of the top drug candidates adds valuable context and biological plausibility.
Strengths:
-
The use of open-access datasets and reproducible tools is commendable.
-
The integration of BiClusO biclustering and disease similarity networks provides a novel angle.
-
The final candidate list and accompanying literature review are highly relevant.
Areas for Improvement:
-
English language and grammar: While the manuscript is understandable, several grammatical issues and awkward phrasings reduce clarity. It would benefit from thorough language editing by a native speaker or professional editor.
-
Methodological clarity: The description of the BiClusO algorithm and DGDA construction should be slightly more detailed for reproducibility—especially the parameters used and justification for threshold choices.
-
Figures and Tables: Ensure all figures (especially Figures 3–6) are clear, high-resolution, and described in sufficient detail in the figure legends. Some figures are hard to interpret as is.
-
Limitations: While the need for experimental validation is acknowledged, the limitations of the datasets (e.g., bias in DGI databases) should also be discussed more thoroughly.
-
Presentation of Results: The transition between methods and results could be made smoother, perhaps with clearer subheadings or visual summaries (e.g., diagrams comparing drug hits from both methods).
Could be improved, but it is not critical.
